# ReverB-SNN:
# Reversing Bit of the Weight and Activation for Spiking Neural Networks

**Yufei Guo   Yuhan Zhang   Zhou Jie   Xiaode Liu   Xin Tong   Yuanpei Chen   Weihang Peng   Zhe Ma**

## Abstract

The Spiking Neural Network (SNN), a biologically inspired neural network infrastructure, has garnered significant attention recently. SNNs utilize binary spike activations for efficient information transmission, replacing multiplications with additions, thereby enhancing energy efficiency. However, binary spike activation maps often fail to capture sufficient data information, resulting in reduced accuracy. To address this challenge, we advocate reversing the bit of the weight and activation for SNNs, called **ReverB-SNN**, inspired by recent findings that highlight greater accuracy degradation from quantizing activations compared to weights. Specifically, our method employs real-valued spike activations alongside binary weights in SNNs. This preserves the event-driven and multiplication-free advantages of standard SNNs while enhancing the information capacity of activations. Additionally, we introduce a trainable factor within binary weights to adaptively learn suitable weight amplitudes during training, thereby increasing network capacity. To maintain efficiency akin to vanilla **ReverB-SNN**, our trainable binary weight SNNs are converted back to standard form using a re-parameterization technique during inference. Extensive experiments across various network architectures and datasets, both static and dynamic, demonstrate that our approach consistently outperforms state-of-the-art methods.

## 1. Introduction

Artificial Neural Networks (ANNs) are currently extensively applied across various fields such as object recognition (He et al., 2016; Ming et al., 2023), object segmentation (Ronneberger et al., 2015), and object tracking (Bewley et al., 2016). However, to achieve enhanced performance, network architectures are becoming increasingly complex (Huang et al., 2017; Devlin et al., 2018). Several techniques have been proposed to tackle this complexity, including quantization (Gong et al., 2019), pruning (He et al., 2017), knowledge distillation (Hinton et al., 2015; Polino et al., 2018; Zhang et al., 2022), and the emergence of spiking neural networks (SNNs) (Maass, 1997; Li et al., 2021a; Xiao et al., 2021; Wang et al., 2022; Bohnstingl et al., 2022; Yu et al., 2025; Guo et al., 2025; Yao et al., 2023; Guo et al., 2023a). SNNs, touted as the next-generation neural networks, reduce energy consumption by emulating brain-like information processing through spike-based communication, which translates weight and activation multiplications into additions, facilitating multiplication-free inference. Moreover, their event-driven computational model demonstrates superior energy efficiency on neuromorphic hardware platforms (Ma et al., 2017; Akopyan et al., 2015; Davies et al., 2018; Pei et al., 2019).

However, it has been observed that SNNs' binary spike activation maps suffer from limited information capacity, failing to adequately capture membrane potential information during quantization, thereby diminishing accuracy (Guo et al., 2022d;a; Wang et al., 2023; Sun et al., 2022). To address this, some studies have explored alternatives such as ternary spikes (Sun et al., 2022), integer spikes (Wang et al., 2023; Fang et al., 2021b; Feng et al., 2022), and even real-valued spikes (Guo et al., 2024c;d; 2025), however these approaches often come at the cost of increased energy consumption due to the inability to convert weight and activation multiplications into additions.

Recent research (Gong et al., 2019; Qin et al., 2024) indicates that using low-bit weights in ANNs can achieve higher accuracy compared to low-bit activations. Motivated by these findings, this paper proposes a novel approach to enhance spike activation's information capacity while preserving the advantages of multiplication-free and event-driven SNNs. Specifically, unlike the conventional binary spike activation approach, we advocate for real-valued spike activations similar to EGRU (Subramoney et al., 2023) to

---

Intelligent Science & Technology Academy of CASIC, China. Yufei Guo and Yuhan Zhang contributed equally to this study. Correspondence to: Yufei Guo <yfguo@pku.edu.cn>, Zhe Ma <mazhe_thu@163.com>.

*Proceedings of the $42^{nd}$ International Conference on Machine Learning*, Vancouver, Canada. PMLR 267, 2025. Copyright 2025 by the author(s).

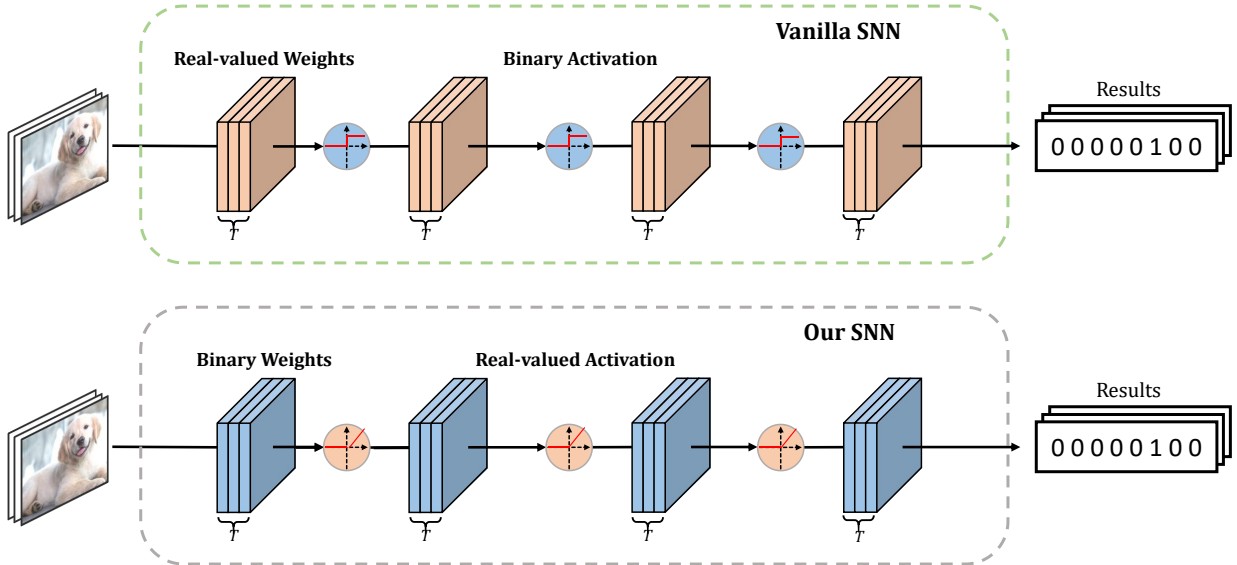

*Figure 1.* The difference between our SNN and the vanilla SNN. Our SNN differs significantly from the vanilla SNN. The vanilla SNN employs binary spikes, leading to significant information loss of the activations. In contrast, our SNN utilizes real-valued spikes alongside binary weights, thereby enhancing the neuron's information capacity. This approach retains the benefits of event-driven processing and multiplication-addition transformations.

increase information capacity. Correspondingly, we adapt real-valued weights to binary weights $\{-1, 1\}$, ensuring retention of multiplication-free and event-driven benefits. Recognizing that binary weights may limit network capacity, we extend them to a learnable form $\{-\alpha, \alpha\}$, where $\alpha$ is a learnable parameter. During inference, we introduce a re-parameterization technique to integrate the $\alpha$ factor into the activation process, thereby preserving the multiplication-free inference capability still.

The distinction between our SNN and the conventional SNN is illustrated in Fig. 1. In summary, our contributions can be summarized as follows:

- We advocate enhancing the information capacity of spike activations by employing real-valued spikes alongside binary weights in SNNs. This approach preserves the multiplication-free and event-driven advantages of standard SNNs while introducing a novel paradigm with real-valued spike neurons and binary weights.

- Additionally, we propose a variant with learnable binary weights and a re-parameterization technique. During training, the weight magnitude $\alpha$ is learned, and during inference, this magnitude is folded into the activation via re-parameterization. This ensures that the binary weights $\{-\alpha, \alpha\}$ revert to standard binary weights $\{-1, 1\}$, maintaining the addition-only advantage.

- We evaluate our methods on both static datasets (CIFAR-10 (Krizhevsky et al., 2010), CIFAR-100 (Krizhevsky et al., 2010), ImageNet (Deng et al., 2009)) and spiking datasets (CIFAR10-DVS (Li et al., 2017)) using widely adopted backbones. Results demonstrate the effectiveness and efficiency of our approach. For instance, using ResNet34 with only 4 timesteps, our method achieves a top-1 accuracy of 70.91% on ImageNet, representing a 3.22% improvement over other state-of-the-art SNN models.

## 2. Related Work

In this section, we provide a brief overview of recent advancements in SNNs focusing on two key aspects: learning methods and information loss reduction techniques.

### 2.1. Learning Methods of Spiking Neural Networks

There are primarily two approaches to achieving high-performance deep SNNs. The first approach involves converting a well-trained ANN into an SNN, known as ANN-SNN conversion (Han & Roy, 2020; Kim et al., 2020; Han et al., 2020; Liu et al., 2022; Yu et al., 2021). This method maps parameters from a pretrained ANN to its SNN counterpart by aligning ANN activation values with SNN average firing rates. Despite its widespread use due to the resource efficiency compared to training SNNs from scratch, ANN-SNN conversion has inherent limitations. It is constrained by rate-coding schemes and overlooks the temporal dynam-

ics unique to SNNs, limiting its efficacy with neuromorphic datasets. Moreover, achieving comparable accuracy to ANNs typically requires numerous timesteps, increasing energy consumption contrary to SNN's low-power design intent. Additionally, SNN accuracy often falls short of ANN accuracy, constraining SNN's potential and research value.

Training SNNs directly from scratch, particularly suited for neuromorphic datasets, has gained attention for its efficiency in reducing timesteps, sometimes to fewer than 5 (Guo et al., 2022d; Fang et al., 2021a; Wu et al., 2018; Rathi & Roy, 2020; Wu et al., 2019; Neftci et al., 2019; Ren et al., 2023). Another emerging approach is hybrid learning, which blends the benefits of ANN-SNN conversion and direct training methods (Rathi & Roy, 2020; Wu et al., 2021a; Zhang et al., 2024; Guo et al., 2023b; 2024d). This approach has also garnered significant interest. In this paper, we focus on enhancing the performance of directly trained SNNs by addressing information loss, an area underexplored in existing literature.

## 2.2. Information Loss Reducing Methods of Spiking Neural Networks

Several methods aim to mitigate information loss of the activation in SNNs by altering spike activation precision (Guo et al., 2022c;a;b; Wang et al., 2023). For instance, a ternary spike neuron transmitting information via $\{0, 1, 2\}$ spikes was proposed in (Sun et al., 2022), which enhances information capacity but lacks the multiplication-addition transformation advantage. Then, a new method was improved upon this with a ternary spike using $\{-1, 0, 1\}$ values, maintaining both improved activation information capacity and the multiplication-addition advantage (Guo et al., 2024b). In MT-SNN (Wang et al., 2023) and DS-ResNet (Feng et al., 2022), a multiple threshold (MT) algorithm was introduced for Leak-Integrate-Fire (LIF) neurons, allowing omission of integer spikes to enhance information transfer. SEWNet (Fang et al., 2021b) proposed an integer spike format by modifying the position of the shortcut module. Some approaches employ real-valued spikes directly to significantly boost information capacity (Guo et al., 2024c; 2025). Nevertheless, these above works all are albeit at the cost of increased energy consumption.

This paper explores the adoption of real-valued spike activation while preserving a multiplication-free advantage using binary weights.

## 3. Methodology

In this section, we begin by introducing the fundamentals of SNN to illustrate its method of information processing and its inherent limitations in information loss. Subsequently, we introduce our **ReverB-SNN** method as a solution to

overcome the challenge. Finally, we propose a variant with learnable binary weights aimed at further enhancing network capacity.

### 3.1. Preliminary

The spike neuron, inspired by the brain's functionality, serves as the fundamental and distinctive computing unit within SNNs. It closely mimics the behavior of biological neurons, characterized by the interplay between membrane potential and input current dynamics. In this paper, we focus on the widely used Leaky-Integrate-and-Fire (LIF) neuron model, which is governed by the iterative equation (Wu et al., 2019):

$$U_l^t = \tau U_l^{t-1} + \mathbf{W}_l O_{l-1}^t, \qquad U_l^t < V_{\text{th}}. \qquad (1)$$

Here, $U_l^t$ represents the membrane potential at time-step $t$ for the $l$-th layer, $O_{l-1}^t$ denotes the spike output from the preceding layer, $\mathbf{W}_l$ denotes the weight matrix at the $l$-th layer, $V_{\text{th}}$ is the firing threshold, and $\tau$ is the time constant governing the leak in the membrane potential. When the membrane potential surpasses the firing threshold, the neuron emits a spike and resets to its resting state, characterized by:

$$O_l^t = \begin{cases} 1, & \text{if } U_l^t \geq V_{\text{th}} \\ 0, & \text{otherwise} \end{cases}. \qquad (2)$$

While the binary spike-based processing paradigm is energy-efficient, it suffers from suboptimal task performance due to limitations in information representation. This motivates our exploration of alternative approaches to enhance the information capacity of SNNs.

**Classifier in SNNs.** In a classification model, the final output is typically processed using the $\mathrm{Softmax}$ function to predict the desired class object. In the context of SNN models, a common approach, as seen in recent studies (Guo et al., 2022c;d; Fang et al., 2021c), involves aggregating the outputs from all time steps to obtain the final output:

$$O_{\text{out}} = \frac{1}{T} \sum_{t=1}^{T} O_{\text{out}}^t. \qquad (3)$$

Subsequently, the cross-entropy loss is computed using the true label and $\mathrm{Softmax}(O_{\text{out}})$.

### 3.2. ReverB: Reversing the Bit of Weight and Activation

To address the issue of information loss in activation, we introduce the **ReverB-SNN** method, inspired by recent research highlighting greater accuracy degradation from quantizing activations compared to weights (Gong et al., 2019; Qin et al., 2024). Specifically, we employ real-valued spike activations, where the output spike at time $t$ for the $l$-th layer is defined as follows:

$$O_l^t = \begin{cases} U_l^t, & \text{if } U_l^t \geq V_{\text{th}} \\ 0, & \text{otherwise} \end{cases}. \qquad (4)$$

Meanwhile, to maintain the multiplication-free and event-driven advantages of standard SNNs, the real-valued weights are converted to binary weights. Consequently, the membrane potential dynamics are updated as:

$$U_l^t = \tau U_l^{t-1} + \mathbf{W}_l^b O_{l-1}^t, \qquad U_l^t < V_{\text{th}}, \qquad (5)$$

where $\mathbf{W}^b = \text{sign}(\mathbf{W}) = \begin{cases} +1, & \text{if } \mathbf{W} \geq 0 \\ -1, & \text{otherwise} \end{cases}$. This approach ensures that activations remain real-valued while leveraging binary weights for computational efficiency, thereby mitigating accuracy loss in SNNs.

**Event-driven Advantage Retaining.** The event-driven signal processing characteristic of SNNs significantly enhances energy efficiency. Specifically, a spiking neuron will only emit a signal and initiate subsequent computations if its membrane potential exceeds the firing threshold, $V_{\text{th}}$; otherwise, it remains inactive. Similarly, our real-valued spike neuron also benefits from this event-driven property. It activates and emits a real-valued spike to initiate computations only when its membrane potential exceeds $V_{\text{th}}$.

**Addition-only Advantage Retaining.** The ability of SNNs to use addition instead of multiplication contributes significantly to their energy efficiency. In a standard binary spike neuron, when a spike is fired, it traditionally multiplies a weight, $w$, connected to another neuron to transmit information, expressed as:

$$x = 1 \times w, \qquad (6)$$

where $w \in \mathbb{R}$. Given the spike amplitude is 1, this multiplication simplifies to an addition:

$$x = 0 + w. \qquad (7)$$

In our real-valued spike neuron, although the spike $o$ is real-valued, the weight $w^b \in \mathbb{B}$ is binary, and the multiplication of $o$ and $w^b$ can be represented as:

$$x = o \times 1, \text{or}, o \times -1. \qquad (8)$$

This too can be simplified to an addition operation:

$$x = 0 + o, \text{or}, 0 - o. \qquad (9)$$

In summary, our proposed method enhances the activation expression capability of SNNs while preserving the event-driven and addition-only advantages of traditional SNNs.

**Surrogate Gradient for Weight Binarization.** In vanilla SNNs, the firing behavior of spiking neurons is non-differentiable, necessitating the use of surrogate gradient (SG) methods in many studies (Rathi & Roy, 2020; Guo et al., 2022c) to address this issue. In our SNN framework, while the firing activity of spiking neurons becomes differentiable, the process of weight binarization poses a non-differentiability challenge. Therefore, similar to other SG

methods used for managing firing activity, we adopt Straight Through Estimator (STE) (Rathi & Roy, 2020; Guo et al., 2022c) gradients to tackle this problem. Mathematically, STE surrogate gradients are defined as:

$$\frac{d\mathbf{W}^b}{d\mathbf{W}} = \begin{cases} 1, & \text{if } -1 \leq \mathbf{W} \leq 1 \\ 0, & \text{otherwise} \end{cases}. \qquad (10)$$

This approach allows us to manage the non-differentiability inherent in weight binarization within our SNN framework effectively.

**Training of Our Method.** In our study, we employ the spatial-temporal backpropagation (STBP) algorithm (Wu et al., 2019) to effectively train our SNN models. STBP treats the SNN as a self-recurrent neural network, facilitating error backpropagation akin to principles used in Convolutional Neural Networks (CNNs). The gradient at layer $l$, derived through the chain rule, is expressed as:

$$\frac{\partial L}{\partial \mathbf{W}_l} = \sum_t \left( \frac{\partial L}{\partial O_l^t} \frac{\partial O_l^t}{\partial U_l^t} + \frac{\partial L}{\partial U_l^{t+1}} \frac{\partial U_l^{t+1}}{\partial U_l^t} \right) \frac{\partial U_l^t}{\partial \mathbf{W}_l^b} \frac{\partial \mathbf{W}_l^b}{\partial \mathbf{W}_l}, \qquad (11)$$

where $\frac{\partial \mathbf{W}_l^b}{\partial \mathbf{W}_l}$ is surrogate gradient for the binarization of the weight in $l$-th layer. This approach enables us to train SNNs effectively by propagating errors through time and across network layers, leveraging the benefits of both temporal and spatial information in neural processing.

### 3.3. Learnable Binary Weight Variant

As mentioned earlier, while real-valued activations increase information capacity, binary weights can decrease network capacity. To address this issue, we extend binary weights to a learnable form, not restricted to $\{-1, 1\}$, but rather $\{-\alpha, \alpha\}$ where $\alpha$ is a learnable parameter defined as:

$$\mathbf{W}_{\text{trainable}}^b = \alpha \cdot \text{sign}(\mathbf{W}) = \begin{cases} +1 \cdot \alpha, & \text{if } \mathbf{W} \geq 0 \\ -1 \cdot \alpha, & \text{otherwise} \end{cases}. \qquad (12)$$

Introducing $\alpha$ allows weights to adapt their amplitude. This parameter $\alpha$ is applied in a channel-wise manner across our SNN models. Consequently, the membrane potential dynamics are adjusted to:

$$U_l^t = \tau U_l^{t-1} + \mathbf{W}_{l,\text{trainable}}^b O_{l-1}^t, \qquad U_l^t < V_{\text{th}}. \qquad (13)$$

Regarding gradients, the gradient of $\mathbf{W}_l$ at the layer $l$ is given by:

$$\frac{\partial L}{\partial \mathbf{W}_l} = \sum_t \left( \frac{\partial L}{\partial O_l^t} \frac{\partial O_l^t}{\partial U_l^t} + \frac{\partial L}{\partial U_l^{t+1}} \frac{\partial U_l^{t+1}}{\partial U_l^t} \right) \frac{\partial U_l^t}{\partial \mathbf{W}_l^b} \frac{\partial \mathbf{W}_l^b}{\partial \mathbf{W}_l}. \qquad (14)$$

While the gradient of $\alpha_l$ at the layer $l$ is:

$$\frac{\partial L}{\partial \alpha_l} = \sum_t \left( \frac{\partial L}{\partial O_l^t} \frac{\partial O_l^t}{\partial U_l^t} + \frac{\partial L}{\partial U_l^{t+1}} \frac{\partial U_l^{t+1}}{\partial U_l^t} \right) \frac{\partial U_l^t}{\partial \alpha_l}. \qquad (15)$$

**Algorithm 1** Training and inference of our SNN.

**Training**

**Input**: An SNN to be trained where the precision of weights and activations was reversed; training dataset; total training iteration: $I_{\text{train}}$.

**Output**: The trained SNN.

1: **for** all $i = 1, 2, \ldots, I_{\text{train}}$ iteration **do**
2:     Get mini-batch training data, $\boldsymbol{x}_{\text{in}}(i)$ and class label, $\boldsymbol{y}(i)$;
3:     Feed the $\boldsymbol{x}_{\text{in}}(i)$ into the SNN and calculate the SNN output, $O_{\text{out}}(i)$ by Eq. 3 ;
4:     Compute classification loss $L_{\text{CE}} = \mathcal{L}_{\text{CE}}(O_{\text{out}}(i), \boldsymbol{y}(i))$;
5:     Calculate the gradient w.r.t. $\mathbf{W}$ by Eq. 14 and the gradient w.r.t. $\alpha$ by Eq. 15;
6:     Update $\mathbf{W}$: ($\mathbf{W} \leftarrow \mathbf{W} - \eta \frac{\partial L}{\partial \mathbf{W}}$) and $\alpha$: ($\alpha \leftarrow \alpha - \eta \frac{\partial L}{\partial \alpha}$) where $\eta$ is learning rate.
7: **end for**

**Re-parameterization**

**Input**: The trained SNN with trainable weights and real-valued spikes ; total layer of SNN: $l$.

**Output**: The re-parameterized trained SNN without normalized binary weight and real-valued spikes.

1: **for** all $i = 1, 2, \ldots, l$ number **do**
2:     Fold the parameters of $\alpha_i$ into $i - 1$ firing function by Eq. 18;
3: **end for**

**Inference**

**Input**: The re-parameterized trained SNN; test dataset; total test iteration: $I_{\text{test}}$.

**Output**: The output.

1: **for** all $i = 1, 2, \ldots, I_{\text{test}}$ iteration **do**
2:     Get mini-batch test data, $\boldsymbol{x}_{\text{in}}(i)$ and class label, $\boldsymbol{y}(i)$ in test dataset;
3:     Feed the $\boldsymbol{x}_{\text{in}}(i)$ into the reparameterized SNN and calculate the SNN output, $O_{\text{out}}(i)$ by Eq. 3 ;
4:     Compare the classification factor $O_{\text{out}}(i)$ and $\boldsymbol{y}(i)$ for classification.
5: **end for**

Since the $\mathbf{W}^b_{\text{trainble}}$ and $O$ are both real-valued in our SNN, using trainable weights introduces the challenge that the multiplication of weight and activation cannot be transformed into an addition, potentially losing the computational efficiency advantages of SNNs. To address this, we propose a training-inference decoupling technique. This method converts different amplitude weights into a normalized binary form during the inference phase through re-parameterization, ensuring retention of the multiplication-free efficiency advantages.

| Dataset | Method | Time-step | Accuracy |
|---|---|---|---|
| CIFAR-10 | Vanilla SNN | 2 | 92.80% |
| | **ReverB** | 2 | **94.14%** |
| | **Learnable variant** | 2 | **94.45%** |
| | Vanilla SNN | 4 | 93.85% |
| | **ReverB** | 4 | **94.55%** |
| | **Learnable variant** | 4 | **94.96%** |
| CIFAR-100 | Vanilla SNN | 2 | 70.18% |
| | **ReverB** | 2 | **72.54%** |
| | **Learnable variant** | 2 | **72.95%** |
| | Vanilla SNN | 4 | 71.77% |
| | **ReverB** | 4 | **72.93%** |
| | **Learnable variant** | 4 | **73.28%** |

*Table 1.* Ablation study for the ternary spike on CIFAR.

**Re-parameterization Technique.** To maintain computational efficiency in SNNs during inference, we propose a re-parameterization technique. Obviously, the Eq.16 can be further written as

$$U_l^t = \tau U_l^{t-1} + \alpha_l \mathbf{W}_l^b O_{l-1}^t, \qquad U_l^t < V_{\text{th}}. \qquad (16)$$

To convert the real-valued weight $W_l^b$ back to binary effectively during inference, we fold the $\alpha$ into the output $O_{l-1}^t$ of the previous layer as a new output, defining $O_{\text{new},l-1}^t = \alpha_l O_{l-1}^t$. This adjustment simplifies Eq.16 to:

$$U_l^t = \tau U_l^{t-1} + \mathbf{W}_l^b O_{\text{new},l-1}^t, \qquad U_l^t < V_{\text{th}}. \qquad (17)$$

Thus the real-valued weight will be converted to the binary weight again. In this way, the output spike at time $t$ for $l - 1$ layer is updated as follows:

$$O_{\text{new},l-1}^t = \begin{cases} \alpha U_{l-1}^t, & \text{if } U_{l-1}^t \geq V_{\text{th}} \\ 0, & \text{otherwise} \end{cases}. \qquad (18)$$

Thereby, the multiplication of the weight and the activation could be converted to addition again in the inference.

In summary, by embedding a learnable factor $\alpha$ into the weight during training, we enhance the network capacity. During inference, we extract this factor from the weight and fold it into the output spike of the previous layer. This approach allows us to maintain the advantages of normalized binary weights and real-valued spikes in the trained SNN, without altering the neuron's update process.

For a detailed outline of the training and inference processes of our SNN, refer to Algorithm 1.

## 4. Experiments

We conducted comprehensive experiments to assess the effectiveness of the proposed **ReverB-SNN** method and

| Dataset | Method | Type | Architecture | Timestep | Accuracy |
|---|---|---|---|---|---|
| CIFAR-10 | SpikeNorm (Sengupta et al., 2019) | ANN2SNN | VGG16 | 2500 | 91.55% |
| | Hybrid-Train (Rathi et al., 2020) | Hybrid training | VGG16 | 200 | 92.02% |
| | TSSL-BP (Zhang & Li, 2020) | SNN training | CIFARNet | 5 | 91.41% |
| | TL (Wu et al., 2021b) | Tandem Learning | CIFARNet | 8 | 89.04% |
| | PTL (Wu et al., 2021c) | Tandem Learning | VGG11 | 16 | 91.24% |
| | PLIF (Fang et al., 2021c) | SNN training | PLIFNet | 8 | 93.50% |
| | DSR (Meng et al., 2022) | SNN training | ResNet18 | 20 | 95.40% |
| | KDSNN (Xu et al., 2023) | SNN training | ResNet18 | 4 | 93.41% |
| | Diet-SNN (Rathi & Roy, 2020) | SNN training | ResNet20 | 5 | 91.78% |
| | | | | 10 | 92.54% |
| | Dspike (Li et al., 2021b) | SNN training | ResNet20 | 2 | 93.13% |
| | | | | 4 | 93.66% |
| | STBP-tdBN (Zheng et al., 2021) | SNN training | ResNet19 | 2 | 92.34% |
| | | | | 4 | 92.92% |
| | TET (Deng et al., 2022) | SNN training | ResNet19 | 2 | 94.16% |
| | | | | 4 | 94.44% |
| | RecDis-SNN (Guo et al., 2022c) | SNN training | ResNet19 | 2 | 93.64% |
| | | | | 4 | 95.53% |
| | Real Spike (Guo et al., 2022d) | SNN training | ResNet19 | 2 | 95.31% |
| | | | | 4 | 95.51% |
| | | | ResNet20 | 4 | 91.89% |
| | **ReverB** | SNN training | ResNet19 | 1 | **95.97%**±0.08 |
| | | | | 2 | **96.39%**±0.11 |
| | | | ResNet20 | 2 | **94.14%**±0.08 |
| | | | | 4 | **94.55%**±0.08 |
| | **Learnable variant** | SNN training | ResNet19 | 1 | **96.22%**±0.12 |
| | | | | 2 | **96.62%**±0.11 |
| | | | ResNet20 | 2 | **94.45%**±0.07 |
| | | | | 4 | **94.96%**±0.10 |
| CIFAR-100 | RMP (Han et al., 2020) | ANN2SNN | ResNet20 | 2048 | 67.82% |
| | Hybrid-Train (Rathi et al., 2020) | Hybrid training | VGG11 | 125 | 67.90% |
| | T2FSNN (Park et al., 2020) | ANN2SNN | VGG16 | 680 | 68.80% |
| | Real Spike (Guo et al., 2022d) | SNN training | ResNet20 | 5 | 66.60% |
| | LTL (Yang et al., 2022) | Tandem Learning | ResNet20 | 31 | 76.08% |
| | Diet-SNN (Rathi & Roy, 2020) | SNN training | ResNet20 | 5 | 64.07% |
| | RecDis-SNN (Guo et al., 2022c) | SNN training | ResNet19 | 4 | 74.10% |
| | Dspike (Li et al., 2021b) | SNN training | ResNet20 | 2 | 71.68% |
| | | | | 4 | 73.35% |
| | TET (Deng et al., 2022) | SNN training | ResNet19 | 2 | 72.87% |
| | | | | 4 | 74.47% |
| | **ReverB** | SNN training | ResNet19 | 1 | **77.62%**±0.10 |
| | | | | 2 | **78.13%**±0.13 |
| | | | ResNet20 | 2 | **72.54%**±0.09 |
| | | | | 4 | **72.93%**±0.12 |
| | **Learnable variant** | SNN training | ResNet19 | 1 | **78.06%**±0.08 |
| | | | | 2 | **78.46%**±0.12 |
| | | | ResNet20 | 2 | **72.95%**±0.11 |
| | | | | 4 | **73.28%**±0.08 |

*Table 2.* Comparison with SoTA methods on CIFAR-10(100).

| Method | Type | Architecture | Timestep | Accuracy |
|---|---|---|---|---|
| STBP-tdBN (Zheng et al., 2021) | SNN training | ResNet34 | 6 | 63.72% |
| TET (Deng et al., 2022) | SNN training | ResNet34 | 6 | 64.79% |
| RecDis-SNN (Guo et al., 2022c) | SNN training | ResNet34 | 6 | 67.33% |
| OTTT (Xiao et al., 2022) | SNN training | ResNet34 | 6 | 65.15% |
| GLIF (Yao et al., 2022) | SNN training | ResNet34 | 4 | 67.52% |
| DSR (Meng et al., 2022) | SNN training | ResNet18 | 50 | 67.74% |
| Ternary spike (Guo et al., 2024b) | SNN training | ResNet34 | 4 | 70.12% |
| SSCL (Zhang et al., 2024) | SNN training | ResNet34 | 4 | 66.78% |
| TAB (Jiang et al.) | SNN training | ResNet34 | 4 | 67.78% |
| MPBN (Guo et al., 2023c) | SNN training | ResNet34 | 4 | 64.71% |
| Shortcut back (Guo et al., 2024a) | SNN training | ResNet34 | 4 | 67.90% |
| Multi-hierarchical model (Hao et al., 2023) | SNN training | ResNet34 | 4 | 69.73% |
| SML (Deng et al., 2023) | SNN training | ResNet34 | 4 | 68.25% |
| Real Spike (Guo et al., 2022d) | SNN training | ResNet18 | 4 | 63.68% |
| | | ResNet34 | 4 | 67.69% |
| SEW ResNet (Fang et al., 2021a) | SNN training | ResNet18 | 4 | 63.18% |
| | | ResNet34 | 4 | 67.04% |
| **ReverB** | SNN training | ResNet18 | 4 | **66.22%**±0.16 |
| | | ResNet34 | 4 | **70.74%**±0.13 |
| **Learnable variant** | SNN training | ResNet18 | 4 | **66.58%**±0.14 |
| | | ResNet34 | 4 | **70.91%**±0.13 |

*Table 3.* Comparison with SoTA methods on ImageNet.

its learnable binary weight variant. Our evaluation included comparisons with several SoTA methods across a range of widely recognized architectures. Specifically, we tested spiking ResNet20 (Rathi & Roy, 2020; Sengupta et al., 2019) and ResNet19 (Zheng et al., 2021) on CIFAR-10(100) (Krizhevsky et al., 2010), ResNet18 (Fang et al., 2021a) and ResNet34 (Fang et al., 2021a) on ImageNet, as well as ResNet20 and ResNet19 on CIFAR10-DVS (Li et al., 2017).

In our work, we used the SGD optimizer to train our models with a momentum of 0.9 and a learning rate of 0.1, which decays to 0 following a cosine schedule. For the CIFAR10(100) and CIFAR-DVS datasets, we trained the models for 400 epochs with a batch size of 128. On ImageNet, we trained for 300 epochs with the same batch size. Data augmentation was performed using only a flip operation. The train and test splits follow the settings provided by the official dataset. The membrane potential decay constant $\tau$ is set to 0.25. In these static datasets, $V_{th}$ is 0 all the time since static datasets can not provide timing information. For neuromorphic datasets, we set it to 0.25.

### 4.1. Ablation Study

We conducted a series of ablation experiments to evaluate the effectiveness of the proposed **ReverB-SNN** method and its learnable binary weight variant on the CIFAR-10 and CIFAR-100 datasets, employing ResNet20 as the backbone

with different timesteps. The results are summarized in Table 1.

The baseline accuracy of vanilla ResNet20 with 2 timesteps reaches 92.80% and 70.18% on CIFAR-10 and CIFAR-100 respectively, consistent with previous studies. Implementing the **ReverB-SNN** method significantly improves performance to 94.14% and 72.54% respectively, marking substantial enhancements of approximately 1.30% and 2.50%. Furthermore, integrating the learnable binary weight variant leads to additional performance gains, resulting in final accuracies of 94.45% for CIFAR-10 and 72.95% for CIFAR-100. These findings underscore the efficacy of our approach in enhancing model performance. When the model is evaluated with 4 timesteps, our method continues to demonstrate its effectiveness. The performance improvements observed with this configuration further validate the robustness and efficacy of the **ReverB-SNN** technique, underscoring its potential for enhancing model accuracy across various settings.

### 4.2. Comparison with SoTA Methods

In this section, we conducted a comparative analysis of our approach against SoTA methods. We present the top-1 accuracy results along with the mean accuracy and standard deviation derived from 3 trials. Our evaluation focused initially on the CIFAR-10 and CIFAR-100 datasets. The summarized results are presented in Table 2. For the CIFAR-

| Method | Type | Architecture | Timestep | Accuracy |
|---|---|---|---|---|
| DSR (Meng et al., 2022) | SNN training | VGG11 | 20 | 77.27% |
| GLIF (Yao et al., 2022) | SNN training | 7B-wideNet | 16 | 78.10% |
| STBP-tdBN (Zheng et al., 2021) | SNN training | ResNet19 | 10 | 67.80% |
| RecDis-SNN (Guo et al., 2022c) | SNN training | ResNet19 | 10 | 72.42% |
| Real Spike (Guo et al., 2022d) | SNN training | ResNet19 | 10 | 72.85% |
| Dspike (Li et al., 2021b) | SNN training | ResNet20 | 10 | 75.40% |
| Spikformer (Zhou et al., 2023) | SNN training | Spikformer | 10 | 78.90% |
| SSCL (Zhang et al., 2024) | SNN training | ResNet19 | 10 | 80.00% |
| **ReverB** | SNN training | ResNet19 | 10 | **80.30%**$\pm$0.20 |
| | | ResNet20 | 10 | **77.80%**$\pm$0.10 |
| **Learnable variant** | SNN training | ResNet19 | 10 | **80.50%**$\pm$0.10 |
| | | ResNet20 | 10 | **78.10%**$\pm$0.10 |

*Table 4.* Comparison with SoTA methods on CIFAR10-DVS.

10 dataset, previous methods achieved peak accuracies of 95.53% using ResNet19 and 93.66% using ResNet20 as their backbone architectures. In contrast, our **ReverB-SNN** method achieves 96.39% and 94.55% respectively, while utilizing fewer timesteps. Furthermore, leveraging learnable binary weights enables our SNN models to attain even higher accuracies. Moving to the CIFAR-100 dataset, our learnable binary weight variant applied to ResNet19 and ResNet20 could achieve superior performance with just 2 timesteps. Our method surpasses leading approaches like TET and RecDis-SNN by approximately 4.0% with ResNet19, despite these methods using 4 timesteps. These experimental findings underscore the efficiency and efficacy of our proposed methodology.

| Method | Accuracy | #Flops | #Sops | Energy |
|---|---|---|---|---|
| Vanilla SNN | 92.80% | 3.54M | 71.20M | 49.73uJ |
| **ReverB** | 94.14% | 3.54M | 74.50M | 49.99uJ |

*Table 5.* Energy estimation.

In our subsequent experiments, we evaluated our approach on the challenging ImageNet dataset, renowned for its complexity compared to CIFAR. Table 3 presents the comparative results. Recent SoTA benchmarks on this dataset include RecDis-SNN (Guo et al., 2022c), GLIF (Yao et al., 2022), DSR (Meng et al., 2022), Real Spike (Guo et al., 2022d), and SEW ResNet (Fang et al., 2021a), achieving accuracies of 67.33%, 67.52%, 67.74%, 67.69%, and 67.04% respectively. Our method achieves significantly higher accuracy, reaching up to 70.91%, a 3.22% improvement over other SoTA SNN models. This notable improvement underscores the effectiveness of our approach for large-scale datasets.

In our final evaluation, we applied our SNN model to the CIFAR10-DVS neuromorphic dataset. Utilizing ResNet19

and ResNet20 as our backbone architectures, our method achieved accuracies of 80.50% and 78.10% respectively, transcending the 80% milestone for ResNet19 even. This marks a substantial improvement in performance on this widely used neuromorphic dataset.

## 5. Energy Estimation

In this section, we evaluate the hardware energy cost associated with the vanilla SNN model and the **ReverB-SNN** model using ResNet20 on the CIFAR10 with 2 timesteps for a single image inference. Since the first rate-encoding layer does not enjoy the multiplication-free property, it will produce the FLOPs (floating point operations). While other layers are calculated by SOPs (synaptic operations). The SOPs are calculated by $s \times T \times A$, where $s$ is the mean sparsity, $T$ is the timestep and $A$ denotes the number of additions in the equivalent artificial neural network (ANN) model. For the Vanilla model, the sparsity of the SNN is 16.42%, whereas for the **ReverB-SNN** model, it is 17.18%. We calculate energy consumption based on the methodology outlined in (Hu et al., 2021), where one FLOP consumes 12.5 pJ and one SOP consumes 77 fJ. A summary of the energy costs is provided in Table 5. The **ReverB-SNN** method results in only a modest 0.52% increase in energy cost compared to the baseline vanilla model. This minimal increase highlights the efficiency of the **ReverB-SNN** approach, demonstrating that it can achieve improved performance with a relatively small additional energy expenditure.

## 6. Conclusion

This study has introduced **ReverB-SNN**, a novel approach for enhancing SNNs by integrating real-valued spike activations with binary weights. Our method addresses the challenge of reduced accuracy in SNNs due to limited information capture by binary spike activation maps. By

reversing the bit of both weights and activations, we have preserved the energy-efficient and multiplication-free characteristics of traditional SNNs while significantly boosting the information capacity of activations. Moreover, the introduction of a trainable factor within binary weights has enabled adaptive learning of weight amplitudes during training, thereby enhancing the overall network capacity. Importantly, to ensure operational efficiency comparable to standard SNNs, we proposed a re-parameterization technique that converts trainable binary weight SNNs back to standard form during inference. Extensive experimental validation across diverse network architectures and datasets, encompassing both static and dynamic scenarios, consistently demonstrates the superiority of our approach over existing state-of-the-art methods.

## Acknowledgements

This work was supported by the National Key Research and Development Program of China (No. 2024YDLN0013) and the National Natural Science Foundation of China (No. 12202412).

## Impact Statement

This paper presents work whose goal is to advance the field of Machine Learning. There are many potential societal consequences of our work, none which we feel must be specifically highlighted here.

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
