# OpenReview forum: "ReverB-SNN: Reversing Bit of the Weight and Activation for Spiking Neural Networks"
_ICML.cc/2025/Conference — ICML 2025 poster_

### Official Review · Reviewer_aSHw · 2025-03-11

**Overall Recommendation:** 4

**Summary:**

This paper introduces a novel binary design in SNN termed ReverB, which uses real-value activation and binary weights, merging the characteristics of both BNN and SNN. This innovative approach retains the energy efficiency advantages of SNNs in inference and their temporal properties. Additionally, the paper proposes a reparameterization method to enhance the performance of the ReverB network.

**Claims And Evidence:**

Yes, the claims made in the submission are supported by clear and convincing evidence. The method is evalued across various network architectures and datasets.

**Essential References Not Discussed:**

I think the paper has cited enough relevant literatures.

**Experimental Designs Or Analyses:**

The experimental design is generally reasonable. The method is evalued across various network architectures and datasets. The authors also conducted a series of ablation experiments to evaluate the effectiveness of the proposed method.

**Methods And Evaluation Criteria:**

The method proposed in this paper is effective. The new network, positioned between BNNs and SNNs, is expected to fully leverage the advantages of both, and potentially start a new field in lightweight networks. The extensive experimental validation across diverse architectures and datasets convincingly demonstrates the superiority of this method over current state-of-the-art approaches.

**Other Comments Or Suggestions:**

None

**Other Strengths And Weaknesses:**

The paper is well-written, the idea is interesting, and the results are impressive.

**Questions For Authors:**

1.Does the ReverB network has a reset mechanism, and how does it function?
2.Does the ReverB network use the Batch Normalization (BN) layer, and can ReverB eliminate it during inference similar to SNN?
3.Why does ReverB exhibit performance advantages over traditional SNNs, and can the authors provide an explanation for this?

**Relation To Broader Scientific Literature:**

This paper introduces a novel binary design in SNN termed ReverB, which uses real-value activation and binary weights, merging the characteristics of both BNN and SNN. It potentially starts a new field in lightweight networks.

**Theoretical Claims:**

Not applicable. This paper does not involve complex theoretical proofs.

---

> ### Author Rebuttal · Authors · 2025-03-31
>
> Thanks for your efforts in reviewing our paper and your recognition of our novel method and effective results. The responses to your questions are given piece by piece as follows.
>
> **Question 1**: Does the ReverB network has a reset mechanism, and how does it function?
>
> **A1**: Thanks for the question. ReverB network has a reset mechanism. When the membrane potential exceeds the firing threshold, it will emit itself and then reset to 0; otherwise, it will decay to a new value and then be updated with next-time input.
>
> **Question 2**: Does the ReverB network use the Batch Normalization (BN) layer, and can ReverB eliminate it during inference similar to SNN?
>
> **A2**: Thanks for the question. We use the BN layer in the network and it can be eliminated in the inference. At inference time, the batch statistics (mean and variance) are no longer needed. Instead, we can fold the normalization, scaling, and shifting operations into the weights and biases of the previous layer. Then $W′=σ/γ⋅W$ and $b′=γ(b−σ/μ)+β$. In our paper, we propose the re-parameterization method to fold the $σ/γ$ into the previous layer activation (see Eq.16). Thus the BN can be removed still in the ReverB.
>
> **Question 3**: Why does ReverB exhibit performance advantages over traditional SNNs, and can the authors provide an explanation for this?
>
> **A3**: Thanks for the question. This is because ReverB quantizes the weight while traditional SNNs quantize activation. Highlight greater accuracy degradation from quantizing activations compared to weights. This is due to several key reasons:
>
> **First**, activations often have a much wider and more varied dynamic range compared to weights. Due to the large dynamic range and possible non-uniform distribution of activations, quantizing them requires compressing a broader set of values into a smaller bit-width, which can result in greater precision loss.
>
> **Second**, Activations change frequently during the forward pass, as they are directly influenced by the input data and the weights. The values can vary significantly between different layers, and this variability can be much greater in deeper layers. Such frequent and complex changes make activations more sensitive to quantization errors. On the other hand, weights are stable and unrelated to input, making them less prone to large errors when quantized.
>
> **Third**, Here, we will also use the entropy concept to show that ReverB-SNN has a higher information capacity than vanilla SNN. thus exhibiting performance advantages over traditional SNNs.
>
> We first perform a theoretical analysis using the concept of entropy. The representational capability $ \mathcal{C}(\mathbf{X}) $ of a set $ \mathbf{X} $ is determined by the maximum entropy of $ \mathbf{X} $, expressed as:
> $\mathcal{C}(\mathbf{X}) = \max \mathcal{H}(\mathbf{X}) = - \sum_{x \in \mathbf{X}} p_{\mathbf{X}}(x) \log p_{\mathbf{X}}(x),$ where $ p_{\mathbf{X}}(x) $ is the probability of a sample $ x $ from $ \mathbf{X} $. Then it is clear for the following proposition:
>
> For a set $ \mathbf{X} $, when the probability distribution of $ \mathbf{X} $ is uniform, i.e., $ p_{\mathbf{X}}(x) = \frac{1}{N} $, where $ N $ is the total number of samples in $ \mathbf{X} $, the entropy $ \mathcal{H}(\mathbf{X}) $ reaches its maximum value of $ \log(N) $. Hence, we conclude that $ \mathcal{C}(\mathbf{X}) = \log(N) $.
>
> Using the proposition, we can evaluate the representational capacity of vanilla SNN and our model.
>
> Let's consider two connected neuron layers. Since the connectivity of the neurons is the same no matter for vanilla SNN or our model, let us focus on an arbitrary two connected neurons from different layers.
>
> For the vanilla SNN, the values output from one neuron to another are {0, 1} x v, where v is the fixed weight between the two neurons. Thus one neuron from vanilla SNN could transform two samples into another one, and $ \mathcal{C}(\mathbf{X}) = \log(2) = 1$.
>
> For our model, the values output from one neuron to another are u x {-1, 1}, where u is the real-valued spike and could be changed. u requires 32-bits. Thus the number of possible samples from u is 2^{32} and the value samples from one neuron to another is 2^{32+1}. $ \mathcal{C}(\mathbf{X}) = \log(2^{32+1}) = 33$.
>
> This highlights the limited representational capacity of vanilla SNN compared to our model.

---

> > ### Comment · Reviewer_aSHw · 2025-04-05
> >
> > Thanks for the author's responses. My concerns have been addressed.

---

### Official Review · Reviewer_LvHJ · 2025-03-14

**Overall Recommendation:** 3

**Summary:**

The paper proposes an SNN design with real-valued activations and binary weights to boost information capacity while keeping energy efficiency. Its novel bit-reversal strategy and adaptive weight scaling are key innovations. However, the paper’s motivation and presentation lack clarity and could benefit from visual aids.

**Claims And Evidence:**

Yes

**Essential References Not Discussed:**

N/A

**Experimental Designs Or Analyses:**

The overall experimental design appears sound and appropriate for the problem at hand. While some aspects could be clarified further, these do not undermine the main findings.

**Methods And Evaluation Criteria:**

Yes

**Other Comments Or Suggestions:**

N/A

**Other Strengths And Weaknesses:**

Strengths
+ The approach preserves the multiplication-free, event-driven nature of SNNs, ensuring that energy efficiency remains a strong point despite the introduction of real-valued activations.
+ Incorporating a trainable factor for binary weights and using re-parameterization during inference allows the network to learn optimal weight magnitudes while still converting to a standard binary format. This offers a good balance between learning flexibility and inference efficiency.

Weaknesses
- The paper does not clearly explain the rationale behind reversing the bits of weight and activation. This lack of clarity could make it challenging for readers to fully grasp why this approach is beneficial.
- The section detailing the contributions is somewhat lengthy and could be streamlined. Consolidating similar ideas might improve readability and focus.
- The comparisons in Table 2 and Table 4 appear to rely on methods from 2022, missing more recent literature.
- The paper could benefit from additional visualizations. Graphs or schematic diagrams illustrating the architecture and the re-parameterization process would enhance understanding and provide clearer insights into the proposed method.

**Questions For Authors:**

1. Could you clarify the theoretical motivation and advantages behind "Reversing Bit"?
2. The paper lacks visualizations; could you add diagrams of the network architecture or re-parameterization process for clarity?

**Relation To Broader Scientific Literature:**

The paper builds on established SNN research, particularly findings that quantizing activations causes more accuracy loss than weights. Its contributions extend prior work on efficient, multiplication-free SNNs and adaptive parameterization.

**Theoretical Claims:**

The paper doesn't provide formal proofs but offers partial derivations that seem generally sound, though some steps lack full justification, leaving a degree of uncertainty.

---

> ### Author Rebuttal · Authors · 2025-03-31
>
> Thanks for your efforts in reviewing our paper and your recognition of our novel bit-reversal strategy and adaptive weight scaling. The responses to your weaknesses and questions are given piece by piece as follows.
>
> **Weakness 1**: The paper does not clearly explain the rationale behind reversing the bits of weight and activation. This lack of clarity could make it challenging for readers to fully grasp why this approach is beneficial.
>
> **R1**: Thanks for the advice. Here, we will use the entropy concept to show that ReverB-SNN has a higher information capacity than vanilla SNN.
>
> we first perform a theoretical analysis using the concept of entropy. The representational capability $ \mathcal{C}(\mathbf{X}) $ of a set $ \mathbf{X} $ is determined by the maximum entropy of $ \mathbf{X} $, expressed as:
> $\mathcal{C}(\mathbf{X}) = \max \mathcal{H}(\mathbf{X}) = - \sum_{x \in \mathbf{X}} p_{\mathbf{X}}(x) \log p_{\mathbf{X}}(x),$ where $ p_{\mathbf{X}}(x) $ is the probability of a sample $ x $ from $ \mathbf{X} $. Then it is clear for the following proposition:
>
> For a set $ \mathbf{X} $, when the probability distribution of $ \mathbf{X} $ is uniform, i.e., $ p_{\mathbf{X}}(x) = \frac{1}{N} $, where $ N $ is the total number of samples in $ \mathbf{X} $, the entropy $ \mathcal{H}(\mathbf{X}) $ reaches its maximum value of $ \log(N) $. Hence, we conclude that $ \mathcal{C}(\mathbf{X}) = \log(N) $.
>
> Using the proposition, we can evaluate the representational capacity of vanilla SNN and our model.
>
> Let's consider two connected neuron layers. Since the connectivity of the neurons is the same no matter for vanilla SNN or our model, let us focus on an arbitrary two connected neurons from different layers.
>
> For the vanilla SNN, the information output from one neuron to another are {0, 1} x W, where W is the fixed weight between the two neurons. Thus one neuron from vanilla SNN could transform two samples into another one, and $ \mathcal{C}(\mathbf{X}) = \log(2) = 1$.
>
> For our model, the information output from one neuron to another are o x {-1, 1}, where o is the real-valued spike and could be changed. o requires 32-bits. Thus the number of possible samples from o is 2^{32} and the value samples from one neuron to another is 2^{32+1}. $ \mathcal{C}(\mathbf{X}) = \log(2^{32+1}) = 33$.
>
> This highlights the limited representational capacity of vanilla SNN compared to our model.
>
> **Weakness 2**: The section detailing the contributions is somewhat lengthy and could be streamlined. Consolidating similar ideas might improve readability and focus.
>
> **R2**: Thanks for the advice. We will further polish our contributions in the final version.
>
> **Weakness 3**: The comparisons in Table 2 and Table 4 appear to rely on methods from 2022, missing more recent literature.
>
> **R3**: Thanks for the advice. We have added more comparisons as below. It can be seen that our method also performs on par with or better than state-of-the-art methods.
>
> | Dateset | Method | Architecture | Timestep | Accuracy |
> | --- | --- | --- | --- | --- |
> | CIFAR10 | Q-SNNs(ACMMM 2024) | ResNet19 | 2 | 95.54% |
> |  | AGMM(AAAI 2025) | ResNet19 | 2 | 96.33% |
> |  | FSTA-SNN(AAAI 2025) | ResNet20 | 4 | 94.72% |
> |  | FSTA-SNN(AAAI 2025) | ResNet19 | 2 | 96.52% |
> |  | TAB(NeurIPS 2024) | ResNet19 | 2 | 94.73% |
> |  | **Our method** | ResNet20 | 4 | **94.96%** |
> |  | **Our method** | ResNet19 | 2 | **96.62%** |
> | CIFAR100 | SSCL(AAAI 2024) | ResNet20 | 2 | 72.86% |
> |  | SSCL(AAAI 2024) | ResNet19 | 2 | **78.79%** |
> |  | TAB(NeurIPS 2024) | ResNet19 | 2 | 76.31% |
> |  | **Our method** | ResNet20 | 4 | **73.28%** |
> |  | **Our method** | ResNet19 | 2 | 78.46% |
> | CIFAR10-DVS | SSCL(AAAI 2024) | ResNet19 | 10 | 80.00% |
> |  | SpikeFormer(ICLR 2023) | SpikeFormer | 10 | 78.90% |
> |  | **Our method** | ResNet19 | 10 | **80.50%** |
>
> **Weakness 4**: The paper could benefit from additional visualizations. Graphs or schematic diagrams illustrating the architecture and the re-parameterization process would enhance understanding and provide clearer insights into the proposed method.
>
> **R4**: Thanks for the advice. We have added the visualizations for the re-parameterization process. Please see it from https://imgur.com/GiJYTke
>
> **Question 1**: Could you clarify the theoretical motivation and advantages behind "Reversing Bit"?
>
> **A1**: Thanks for the question.  Please see our response to **Weakness 1.**
>
> **Question 2**: The paper lacks visualizations; could you add diagrams of the network architecture or re-parameterization process for clarity?
>
> **A1**: Thanks for the advice. We have added the visualizations for the re-parameterization process. Please see it from https://imgur.com/GiJYTke

---

> > ### Comment · Reviewer_LvHJ · 2025-04-06
> >
> > The authors have addressed most of my concerns, which makes me relatively satisfied with the revisions. I lean towards a weak accept. However, the paper still requires further improvements in several areas:
> >
> > My primary interest was in the visualization of experimental results rather than inference visualization. This focus might not have been clearly communicated in the submission.
> >
> > The qualitative and quantitative comparison methods could be enhanced by including more recent work, such as [a] Towards Low-latency Event-based Visual Recognition with Hybrid Step-wise Distillation Spiking Neural Networks.
> >
> > There are some formatting and typesetting issues that need to be corrected.

---

### Official Review · Reviewer_jZSA · 2025-03-14

**Overall Recommendation:** 2

**Summary:**

This paper addresses the issue of information loss in Spiking Neural Networks (SNNs) due to the binarization of activations. The main contribution in the paper is to use binary weights (in $\\{-1,1\\}$) and real-valued spikes, instead of binary spikes and real-valued weights. This initial contribution is extended by allowing the binary weights to take their values in $\\{-\alpha, \alpha\\}$, where $\alpha$ is a trainable parameter; then, a re-parametrization technique is proposed to take $\alpha$ into account at inference while going back to binary weights in $\\{-1,1\\}$. Experiments on three datasets (CIFAR-10, CIFAR-100, ImageNet, with two architectures tested per dataset) are reported. Results suggest that the proposed contributions improve the classification accuracy over the baseline architectures, and perform on par with or better than state-of-the-art architectures.

## Update after rebuttal
The rebuttal of the authors addressed my concerns about the rationale and design of the contribution. However, I still have concerns about the way hyperparameters were chosen/optimized, and the validity of the model used to estimate energy consumption. So, I increase my score to _2. Weak reject_.

**Claims And Evidence:**

The paper makes the following claims:
1. Weight binarization is less detrimental to performance than activation binarization. This statement seems to be supported by experimental results.
2. The proposed method (binarized weights, non-binarized spikes) maintains the event-based nature of the network. This claim seems correct, by construction.
3. The proposed method only requires additions. It seems to only apply to inference, not training, although this is no explicitly stated in the paper. While this claim is true for the first version of ReverB, it looks like this is not the case for the learnable version, which requires multiplications, according to Equation 18.

**Essential References Not Discussed:**

I did not identify any essential references that are not cited.

**Experimental Designs Or Analyses:**

I reviewed Section 4 entirely. My comments about the experiments are detailed below.

1. The firing threshold $V_{\mathrm{th}}$ is said to be set initially to 0. Is this the actual value or a typo? Also, can it change over time? This was not mentioned previously in the paper.

2. A number of elements are missing from the experimental settings, which prevents from reproducing the experiments:
    - the optimizer used to train the network and its hyperparameters (e.g., number of epochs),
    - the protocol used to determine hyperparameters,
    - data pre-processing (if any),
    - preparation of the data (train/validation/test splits, and the size of mini-batches).

3. In Table 5, the figures for learnable ReverB are missing.

**Methods And Evaluation Criteria:**

1. The paper proposes to solve the issue of information loss due to binary spikes in SNNs by binarizing weights instead of spikes. On the one hand, I find it interesting to change the perspective on this issue by moving the quantization issue from one variable of the model to another. On the other hand, I wonder whether this approach could be applied in practice. SNNs are valuable when they can be implemented on low-power neuromorphic hardware. Low power consumption is made possible thanks to the use of binary spikes by the model, and neuromorphic hardware is designed with this in mind. Switching binarization from activations to weights may not have the same benefits. This question is not addressed in the paper.

2. The evaluation criteria are the accuracy of the model and its energy consumption, which are relevant criteria in this context. However, the evaluation of the energy consumption of the model is based on the method from (Hu et al., 2021), who base their estimation on the specifications of particular hardware devices. These devices may not be able to run natively networks with real-valued spikes like the ones proposed in this paper. The relevance of this methodology for the network described here is not demonstrated in the paper. In addition, I believe memory consumption should also be considered as an evaluation criteria, as, here again, it may be very different with the proposed model.

**Other Comments Or Suggestions:**

1. Figure 1 is not very informative, as the principle of the contribution is simple enough and clearly stated in the paper. This figure could be removed.

2. In Equation 3, $T$ is not defined.

3. (Rathi & Roy, 2020) is cited to motivate the choice of the surrogate gradient, however a different surrogate gradient function is used in that paper.

4. In Section 3.2, paragraphs "Event-based Advantage Retaining" and "Addition-only Advantage Retaining" could be significantly shortened as they state straightforward properties of their model.

5. Parentheses are not typeset correctly in several equations (`\left(` and `\right)` should be used).

6. Before Equation 16, I think the equation number in "Eq. 16 can be further written as" is not the right one.

7. In Section 4.2, it is stated that top-1 accuracy and the mean accuracy and standard deviation are presented. I guess what is meant is that mean (top-1) accuracy is reported?

8. Percentages (%) are used instead of percentage points (pp) when presenting differences in accuracies.

9. Some typos should be corrected:
    - "trainable" is misspelled "trainble" in equations,
    - "binarization" is misspelled "binaration" after Equation 11,
    - "will converted": "will be converted",
    - before Equation 17, I guess $\alpha_1$ should actually be $\alpha_l$,
    - in the description of the experimental settings (Section 4), $\tau$ becomes $\tau_{\mathrm{decay}}$,
    - "peak accuracies of 95.51%": "peak accuracies of 95.53%",
    - "does not enjoy the multiplication-free" needs to be rephrased (Section 5),
    - "obviously" -> "Obviously",
    - the title of Section 4 should be spelled "Experiments".

10. The paper contains some subjective over-statements, which should be avoided. For instance:
    - "the well-trained SNN" (in Algorithm 1),
    - "Remarkably, our method achieves [...]",
    - "achieved impressive accuracies".

**Other Strengths And Weaknesses:**

1. After Equation 12, it is stated that $\alpha \in \mathbb{R}^{C \times 1 \times 1}$. This dimension is not quite clear to me, since I assumed from the beginning that the previous equations were about fully-connected layers. It seems that the dimensions of tensors do not match here. The dimensions of tensors should be provided to make this clearer.

2. Some elements in Algorithm 1 are not clear.
   - It is not clear whether the `for` loop in the training algorithm loops over mini-batches or epochs.
   - In line 2 of the re-parameterization algorithm, why does $\alpha_i$ unfold into $i-1$ functions?
   - Why are labels used for inference? They should be used for evaluation only, not inference.

**Questions For Authors:**

1. Why considering binary weight in $\\{-1,1\\}$ and not ternary weights in $\\{-1,0,1\\}$? The latter could enhance the learning capacity of neurons.

2. SNNs are valuable when they can be deployed on low-power neuromorphic hardware. Such hardware is typically designed with binary spikes in mind. Can the authors elaborate on the compatibility of their approach with current neuromorphic hardware, and what changes (if needed) should be applied to neuromorphic architectures to run this type of model?

**Relation To Broader Scientific Literature:**

1. The problem addressed in this paper is relevant to the community of neuromorphic machine learning. It has been addressed in a number of previous papers, as mentioned in Section 2. To the best of my knowledge, there is no previous work that proposes the same contribution as the one in this paper.

2. The use of binary weights has been explored in standard ANNs, for instance in (Courbariaux et al., 2015). The paper does not mention this line of work.

- (Courbariaux et al., 2015) M. Courbariaux, Y. Bengio, J.P. David. BinaryConnect: Training Deep Neural Networks with binary weights during propagations. Neural Information Processing Systems (NeurIPS) (2015).

**Theoretical Claims:**

1. The paper provides the formulas for gradient computation in the proposed models (Equations 11, 14, and 15), but do not provide the derivation of these results, so I could not check whether they are sound.
The authors should provide (for instance, as supplementary material) the complete demonstration that leads to these equations, as it is not straightforward considering the changes they made to the initial model (continuous weights, binary spikes) used in the STBP paper (Wu et al, 2018).

2. Equations 14 and 11 are the same, whereas Equation 14 should show how to compute the gradients with $\mathbf{W}^b_{\mathrm{trainable}}$. I believe there might be an error here.

3. The authors state (Section 3.2) that "the firing activity of spiking neurons becomes differentiable" in their model. This statement should be further justified. To my understanding, although the spikes take real values, they are still local in time and generated through a thresholding function, so there are discontinuities in the activation function that should be problematic in terms of differentiability.

---

> ### Author Rebuttal · Authors · 2025-03-31
>
> Thanks for your efforts in reviewing our paper. We will try to make the work clearer for you. The responses to your concerns and questions are given as follows.
>
> **Concern 1**: The only required addition is not for the learnable version.
>
> **R1**: Sorry for the confusion. Since the $\alpha$ will be fixed after training, it can be folded to the activation function before inference. Then only requires additions too.
>
> **Concern 2**: the model may not have the low power consumption benefits.
>
> **R2**: Thanks for the question. I agree that on many low-power neuromorphic hardware, the low-power consumption is thanks to the use of binary spikes. However, there are also many hardware that realize low-power consumption based on replacing multiplications with additions like [1,2]. With this hardware, our model can keep low power consumption too. What’s more, there is also no available hardware that could support SNN-based transformer architectures which become popular now. Our model and SNN-based transformer architectures could drive further hardware development like the emergence of other hardware like Lohoi and TureNorth.
>
> [1] A systolic SNN inference accelerator and its co-optimized software framework.
>
> [2] Tianjin chip.
>
> **Concern 3**: The evaluation of memory consumption.
>
> **R3**: Thanks for the question. For memory consumption, the vanilla SNN’s weights adopt 32 bits while our model’s weights could adopt 1 bit. Thus the memory consumption of our model is much less than that of the vanilla SNN.
>
> **Concern 4**: Provide the derivation for Equations 11, 14, and 15.
>
> **R4**: Thanks for the question. For Equations 11, from Equation 5 and $\mathbf{W_l}^b = {\rm sign}(\mathbf{W_l})$, based on chain rule, we could know that $\frac{\partial {L}}{\partial {\mathbf{W}_l}} = \sum_t \frac{\partial {L}}{\partial {U^t_l}}\frac{\partial {U^t_l}}{\partial {\mathbf{W}^b_l}}\frac{\partial \mathbf{W}^b_l}{\partial {\mathbf{W}_l}}$. From Equation 4 and Equation 1, we know that $U^t_l$ will affect $O^t_l$ and $U^{t+1}_l$, thus $\frac{\partial {L}}{\partial {U^t_l}} = \frac{\partial {L}}{\partial {O^t_l}} \frac{\partial {{O^t_l}}}{\partial {{U^t_l}}} +  \frac{\partial {L}}{\partial {{{U^{t+1}_l}}}} \frac{\partial {{U^{t+1}_l}}}{\partial {{U^t_l}}}$. Combine all these, we can get Equations 11.
>
> The proof of Equation 14 is the same as Equation 11.
>
> For Equation  15, the $\alpha$ only affects $U^t$, thus based on the derivation of Equation 11, it is easy to derive it.
>
> **Concern 5**: Equations 14 and 11 are the same?
>
> **R5**: Sorry for the confusion.  $\mathbf{W}^b_{\rm trainble} = \alpha \mathbf{W}^b$, and we calculate $\alpha$ and $W$ separately in Equations 14 and 15. Thus Equations 14 and 11 are the same in form.
>
> **Concern 6**: The firing activity becomes differentiable" should be further justified.
>
> **R6**: Thanks for the advice. Compared to binary activation, our real-valued activation becomes differentiable in more intervals. We overclaimed this in the paper. We will correct this in the final version. Thanks.
>
> **Concern 7**: The firing threshold is said to be set to 0?
>
> **R7**: Sorry for the confusion. In these static datasets, it is 0 all the time, since static datasets can not provide timing information. For neuromorphic datasets, we set it to 0.25.
>
> **Concern 8**: the experimental settings are missing.
>
> **R8**: Thanks for the question. We will clarify the code settings in detail in the final version.
>
> **Concern 9**: In Table 5, the figures for learnable ReverB are missing.
>
> **R9**: Thanks for the question.  We add the results for learnable ReverB below.
>
> | Accuracy | #Flops |  #Sops | Energy |
> | --- | --- | --- | --- |
> | 94.45% | 3.54M |  75.10M | 50.03uJ |
>
> **Concern 10**: Why $\alpha \in \mathbb{R}^{C \times 1 \times 1}$
>
> **R10**: Sorry for the confusion. $\alpha$ is in a channel-wise manner for convolution layers.
>
> **Concern 11**: Some elements in Algorithm 1 are not clear.
>
> **R11**: Sorry for the confusion. The `for` loop is over mini-batches. With the $\alpha$ folded to i-1 function, then we can obtain a standard ReverB-SNN. We will correct the Inference to Evaluation in the Algorithm. Thanks.
>
> **Concern 12**: Other Comments Or Suggestions.
>
> **R12**: Very much thanks for these kind reminders. We will carefully correct these in our final version.
>
> **Question 1**: Why considering binary weights not ternary weights.
>
> **A1**: Thanks for the question. Using ternary weights is better than binary weights in our experiments. However, we only report the binary weight and real-valued activate results for a fair comparison of vanilla real-valued weight and binary activate SNNs.
>
> **Question 2**: How to deploy the proposed model to neuromorphic hardware.
>
> **A2**: Thanks for the question. Please see our response for **Concern 2**.

---

> > ### Comment · Reviewer_jZSA · 2025-04-03
> >
> > Thanks to the authors for their answers.
> >
> > The rebuttal addresses my concerns only partially. Especially, some elements in the theoretical development (R5, R6, R10) and experimental design (R3 + concerns with energy consumption model, R8) are still unclear. So, I have to maintain my initial score.

---

> > > ### Author Response · Authors · 2025-04-04
> > >
> > > Considering that the first reply is only allowed to be 5000 characters, some issues were not explained in depth. Here, we provide further detailed responses.
> > >
> > > **Concern 5**: Eq 14 and 11 are the same?
> > >
> > > **R5**: Thanks.
> > > Similarly to the derivation of Eq 11 in R4, we can derive that $\frac{\partial {L}}{\partial {\mathbf{W}_{l,trainable}}} = \sum_t (\frac{\partial {L}}{\partial {O^t_l}} \frac{\partial {{O^t_l}}}{\partial {{U^t_l}}} +  \frac{\partial {L}}{\partial {{{U^{t+1}_l}}}} \frac{\partial {{U^{t+1}_l}}}{\partial {{U^t_l}}} )\frac{\partial {{U^t_l}}}{\partial {\mathbf{W}\_{l,trainable}}}$.
> > >
> > > We also have $\mathbf{W}^b_{l,trainable} = \alpha {W^b_l}=\alpha \cdot {\rm sign}(\mathbf{W_l})$, thus $\frac{\partial {L}}{\partial {\mathbf{W}_{l}}} = \frac{\partial {L}}{\partial {\mathbf{W}\_{l,trainable}}} \frac{\partial {\mathbf{W}\_{l,trainable}}}{\partial {\mathbf{W}^b\_l}}\frac{\partial \mathbf{W}^b_l}{\partial {\mathbf{W}\_l}}$.
> > >
> > > Combine the two Equations, we have $\frac{\partial {L}}{\partial {\mathbf{W}_{l}}} = \sum_t (\frac{\partial {L}}{\partial {O^t_l}} \frac{\partial {{O^t_l}}}{\partial {{U^t_l}}} +  \frac{\partial {L}}{\partial {{{U^{t+1}_l}}}} \frac{\partial {{U^{t+1}_l}}}{\partial {{U^t_l}}} )\frac{\partial {{U^t_l}}}{\partial {\mathbf{W}\_{l,trainable}}}\frac{\partial {\mathbf{W}\_{l,trainable}}}{\partial {\mathbf{W}^b\_l}}\frac{\partial \mathbf{W}^b\_l}{\partial {\mathbf{W}\_l}}$.
> > >
> > > Then fold $\frac{\partial {{U^t_l}}}{\partial {\mathbf{W}\_{l,trainable}}}\frac{\partial {\mathbf{W}\_{l,trainable}}}{\partial {\mathbf{W}^b\_l}}$ as $\frac{\partial {{U^t_l}}}{\partial {\mathbf{W}^b\_l}}$, we have Eq 14.
> > >
> > > **Concern 6**: The firing activity becomes differentiable" should be justified.
> > >
> > > **R6**: Sorry for the confusion. Compared to binary activation where the gradient is infinite at $V_{th}$, otherwise 0, our real-valued activation behaves similarly to a ReLU function, as shown in Equation 4. I agree that it remains non-differentiable when $U<V_{th}$, akin to how ReLU is non-differentiable for $x<0$. however,for $U>V_{th}$, the function is differentiable, similar to the behavior of ReLU for $x>0$. In the context of ANNs, ReLU is generally regarded as differentiable, so we stated that our real-valued activation could similarly be considered differentiable. I recognize that this statement may still be somewhat imprecise, and we will make the necessary corrections in the final version.
> > >
> > > **Concern 10**: Why $\alpha \in \mathbb{R}^{C \times 1 \times 1}$
> > >
> > > **R10**: Sorry for the confusion. In the work, we adopt CNN models and the $\alpha$ is set to be learnable for convolution layers. For convolution layers, the $W \in \mathbb{R}^{C_{out} \times C_{in} \times K \times K}$. We set $\alpha$ in a $C_{in}$ channel-wise manner as$\alpha \in \mathbb{R}^{1 \times C_{in} \times 1 \times 1} $, thus it can be folded to the previous activation layers. we mistakenly wrote$\alpha \in \mathbb{R}^{1 \times C \times 1 \times 1}$  as $\alpha \in \mathbb{R}^{C \times 1 \times 1}$, we will correct it in the final version.
> > >
> > > **Concern 3**: The evaluation of energy and memory consumption.
> > >
> > > **R3**: Thanks. I agree that the hardware from (Hu et al., 2021) may not support our models, which makes the theoretical energy consumption somewhat imprecise. However, it is challenging to run the SNN model on hardware to directly compute energy consumption, as model development is often detached from hardware capabilities. Given this limitation, we have adopted the general theoretical energy consumption approach, which is consistent with other works presented in top conferences like ternary spike model [1] and SNN-based transformer model [2] which also cannot run on existing hardware. These works similarly rely on theoretical evaluation methods, like ours, to show their advantages until specialized hardware becomes available.
> > >
> > > [1] Ternary Spike, AAAI 2024
> > >
> > > [2] SPIKE-DRIVEN TRANSFORMER V2, ICLR 2024
> > >
> > > For memory consumption, the vanilla SNN uses full precision weights, requiring 32 bits per weight, whereas our model uses binary weights, requiring only 1 bit per weight. Taking ResNet20 as an example, which has 11.25M parameters, the vanilla SNN model would require $11.25 \times 4 = 45$MB of memory, while our SNN model only requires 1.41 MB of memory.
> > >
> > > **Concern 8**:the experimental settings are missing.
> > >
> > > **R8**: Thank you for the question. We used the SGD optimizer to train our models with a momentum of 0.9 and a learning rate of 0.1, which decays to 0 following a cosine schedule. For the CIFAR10(100) and CIFAR-DVS datasets, we trained the models for 400 epochs with a batch size of 128. On ImageNet, we trained for 300 epochs with the same batch size. Data augmentation was performed using only a flip operation. The train and test splits follow the settings provided by the official dataset.  In these static datasets, $V_{th}$ is 0 all the time since static datasets can not provide timing information. For neuromorphic datasets, we set it to 0.25.

---

### Official Review · Reviewer_LZsQ · 2025-03-19

**Overall Recommendation:** 4

**Summary:**

In this paper, the authors propose to make the weights of a spiking neural network ternary and the spikes of the units in the network real valued, essentially swapping what is done in SNNs usually. This swap preserves the advantages of SNNs, while improving its expressivity. The authors demonstrate their method in a number of benchmarks.

**Claims And Evidence:**

The claims made in the paper are supported by clear and convincing evidence -- both for their method preserving the advantages of SNNs in terms of energy efficiency (Table 5) and its ability to outperform existing SNNs in various benchmarks (Table 2-4).

**Essential References Not Discussed:**

The use of real-valued activations in spiking (i.e. event-based) networks has been done before in [1], and should be discussed. But the specific combination of ternary weights and real-valued activation is novel to my knowledge.

[1] Subramoney, A., Nazeer, K. K., Schöne, M., Mayr, C. & Kappel, D. Efficient recurrent architectures through activity sparsity and sparse back-propagation through time. in The Eleventh International Conference on Learning Representations (2023).

**Experimental Designs Or Analyses:**

The ablation studies, evaluation on standard benchmarks, and energy efficiency are the analyses that the authors perform. Did not find any major issues with any of them.

**Methods And Evaluation Criteria:**

The benchmarks in which the method is evaluated on are pretty standard in the field for feed-forward SNNs. But it would have been useful to also see evaluations on sequence tasks since SNNs are inherently recurrent.

It is not completely clear why the authors choose to use STBP to train the network rather than standard BPTT. This decision could be explained a bit better.

**Other Comments Or Suggestions:**

- It's not clearly described if a value of $\alpha$ is shared across an entire layer.

**Other Strengths And Weaknesses:**

The key concept in this paper is novel to my knowledge. The end to end training setup and evaluation on challenging benchmarks is also a strength.

One minor weakness is the framing of the method within current literature, which is not done thoroughly. For example many of the papers referenced in the introduction seem very arbitrary or is missing references to the seminal papers. E.g. The usual reference used for SNNs is [1]. Quantization was known well before 2019. Ditto knowledge distillation etc.

[1] Maass, W. Networks of spiking neurons: The third generation of neural network models. Neural Networks 10, 1659–1671 (1997).

**Questions For Authors:**

- What is the activation used in between layers? In Fig. 1, it looks like it uses ReLU activations?

**Relation To Broader Scientific Literature:**

This paper builds on existing SNNs, improving their expressivity while keeping their other advantages. The use of ternary weights and real valued spikes [1] have been done before, but not in this specific context.

[1] Subramoney, A., Nazeer, K. K., Schöne, M., Mayr, C. & Kappel, D. Efficient recurrent architectures through activity sparsity and sparse back-propagation through time. in The Eleventh International Conference on Learning Representations (2023).

**Theoretical Claims:**

No major theoretical claims.

---

> ### Author Rebuttal · Authors · 2025-03-31
>
> Thanks for your efforts in reviewing our paper and your recognition of our novel method and notable results. The responses to your concerns and questions are given piece by piece as follows.
>
> **Concern 1**: It would have been useful to also see evaluations on sequence tasks since SNNs are inherently recurrent.
>
> **R1**: Thanks for the advice. Our method also performs well in sequence tasks. We have added the result on the CIFAR10-DVS and DVS-Gesture as below. It can be seen that our method also performs on par with or better than state-of-the-art methods.
>
> | Dateset | Method | Architecture | Timestep | Accuracy |
> | --- | --- | --- | --- | --- |
> | CIFAR10-DVS | SSCL(AAAI 2024) | ResNet19 | 10 | 80.00% |
> | CIFAR10-DVS | SpikeFormer(ICLR 2023) | SpikeFormer | 10 | 78.90% |
> | CIFAR10-DVS | **Our method** | ResNet19 | 10 | **80.50%** |
> | DVS-Gesture | ASA-SNN(ICCV 2023) | 5 layer SCNN | 20 | 97.70% |
> | DVS-Gesture | SpikeFormer(ICLR 2023) | SpikeFormer | 20 | 96.90% |
> | DVS-Gesture | TCJA (TNNLS 2024) | 5 layer SCNN | 20 | 97.56% |
> | DVS-Gesture | **Our method** | 5 layer SCNN | 20 | **98.23%** |
>
> **Concern 2**: Why do the authors choose to use STBP to train the network rather than standard BPTT?
>
> **R2**: Sorry for the confusion.  Considering the similarity in computational mechanisms between SNNs and Recurrent Neural Networks (RNNs), SNN researchers transferred the Back-propagation Through Time (BPTT) method from RNNs to the supervised learning field of SNNs, which is also called the STBP training algorithm. Thus BPTT is the same as STPB in SNNs.
>
> **Concern 3**: One minor weakness is the framing of the method within the current literature.
>
> **R3**: Thanks for the advice. We will add more related literature and reframe them in the final version, such as:
>
> [1] Subramoney, A., Nazeer, K. K., Schöne, M., Mayr, C. & Kappel, D. Efficient recurrent architectures through activity sparsity and sparse back-propagation through time. in The Eleventh International Conference on Learning Representations (2023).
>
> [2] Maass, W. Networks of spiking neurons: The third generation of neural network models. Neural Networks 10, 1659–1671 (1997).
>
> **Concern 4**: It's not clearly described if a value of $\alpha$ is shared across an entire layer.
>
> **R4**: Sorry for the confusion.  $\alpha$ is not shared across an entire layer. It is applied in a channel-wise manner across our models. We will make this description clearer in the final version.
>
> **Questions  1**: What is the activation used in between layers?
>
> **A1**: Sorry for the confusion. In addition inherently recurrent of our activation, the output of our activation is a ReLU-like function defined as follows:
> $ {\rm O}= {\rm U} \\  {\rm if U} \\ \ge V_{\rm th}, \\ { \rm otherwise} \\ 0$
>
> In Relu activation, the $V_{\rm th}$ is 0, while in our activation, the $V_{\rm th}$ can be adjusted across different datasets or scenes. Our activation will decay through time, while ReLU not.

---

### Decision · Program_Chairs · 2025-05-01

**Decision:**

Accept (poster)

**Comment:**

This paper proposes a novel spiking neural network (SNN) paradigm that inverts the conventional quantization approach by employing ternary weights and real-valued spikes, significantly enhancing network expressiveness while preserving energy efficiency. The method demonstrates competitive classification accuracy across various architectures (ResNet, VGG) and datasets (CIFAR, ImageNet), with ablation studies validating its design. While reviewers raised concerns regarding hardware compatibility analysis, hyperparameter optimization transparency, and missing gradient derivations, the core contribution—enhancing SNN expressivity through weight-spike role reversal—remains valid. The authors have addressed major concerns during rebuttal, and remaining issues (e.g., energy evaluation methodology, literature positioning) can be resolved through revisions. Therefore, an acceptance is recommended.